# Electrostatically Driven Vertical Combinatorial Patterning of Colloidal Nano-Objects

**Gaëtan Petit [1], Romain Hernandez [2], Simon Raffy [1], Aurélien Cuche [2], Lorena Soria Marina [3], Michele D'Amico [3], Etienne Palleau [1],\* and Laurence Ressier [1]**

1 Laboratory of Physics and Chemistry of Nano-Objects (LPCNO), Université de Toulouse, CNRS, INSA, UPS, 135 Avenue de Rangueil, 31077 Toulouse, France
2 CEMES-CNRS, Université de Toulouse, 29 Rue Jeanne Marvig, 31055 Toulouse, France
3 NEXDOT, 102 Avenue Gaston Roussel, 93230 Romainville, France
\* Correspondence: epalleau@insa-toulouse.fr; Tel.: +33-561559672

**Abstract:** The hierarchically directed assembly of multiple types of colloidal nano-objects on surfaces is of interest for developing disruptive applications combining their original properties. We propose herein a versatile, electrostatically driven strategy to arrange various kinds of colloids vertically in the shape of 3D micropatterns by nanoxerography. We made the proof of concept of this vertical combinatorial nano-object patterning using two types of photoluminescent CdSe(S)/CdZnS core/shell nanoplatelets emitting in the red and green wavelengths as model colloidal nanoparticles. The key experimental parameters were investigated to tune the thickness of each independent level of nanoplatelets within the vertical stack. We finally applied such a concept to make dual-colored nanoplatelet patterns. Interestingly, we proved numerically that the relatively high index of the nanoplatelet level is responsible for the partially directed emissions observed in photoluminescence experiments.

**Keywords:** directed assembly; quantum nanoplatelets; nanoxerography; combinatorial assembly

## 1. Introduction

The directed assembly of colloidal nanoparticles on surfaces is a prerequisite for studying their original properties and for making the most of them as the active part of functional devices [1–3]. In this context, the precise assembly of different types of colloids to make micropatterns of desired geometry on the same substrate, also referred to as combinatorial nanoparticle patterning, co-, or hierarchical assembly [3,4], has brought about interesting developments. These include the creation of red-green-blue matrices for light-emitting devices [5–7], shadow masks for colloidal lithography [8], and peptide arrays [9], to name a few. This multiple deposition is either performed with various nano-object (NO) patterns arranged horizontally, side by side on the same plane, or by stacking them vertically, generating different layers/levels. Several patterning techniques are available to create them: a chip-based method [10], transfer printing [11], inkjet printing [12], capillary-based techniques [13], external field (electrical or magnetic)-driven assembly [9,14], etc. Among them, the layer-by-layer approach based on electrostatic interactions has been intensively studied to fabricate vertical combinatorial NO films [15] and coupled to lithography technologies [16,17] or topographical structures or electrodes [18,19] to shape them into specific patterns.

In this paper, we propose to use an alternative electrically driven directed assembly technique called nanoxerography to create vertical combinatorial NO patterns. This versatile approach employs charged patterns of desired geometries written on electret surfaces to selectively trap charged and/or polarizable NOs [20]. The combinatorial NO patterning by nanoxerography has been previously demonstrated in horizontal configurations [21,22] but its capabilities in vertical ones have not been investigated yet to the best of our knowledge.

This work serves as a proof of concept for vertical combinatorial NO patterns by nanoxerography using two types of highly photoluminescent CdSe(S)/CdZnS core/shell nanoplatelets as model colloidal nanoparticles and demonstrates how they have been applied to make dual-colored patterns.

## 2. Materials and Methods

### 2.1. Vertical Combinatorial Patterning by AFM Nanoxerography

The vertical combinatorial electrostatically directed assembly of colloidal nano-objects on surfaces presented in this work was performed by atomic force microscopy (AFM) nanoxerography [21,23]. Figure 1 illustrates the three main steps of the proposed concept. The initial injection step consists in writing charged patterns of desired geometries on an electret surface using a polarized AFM (ICON from Bruker, Germany) tip (1). Then the charged substrate is put in contact with a first colloidal dispersion of interest for a few seconds before being naturally dried. During this first development step, NOs attracted by the electrostatic forces generated by the charged patterns, are selectively assembled on them, forming a multilayered microstructure (2). Then the same sample is put in contact again for a few seconds with a second colloidal dispersion, which creates a second level of NOs selectively deposited on top of the first one (3). The main point is to stop the first assembly building quite quickly to prevent NOs from reaching the highest possible assembly thickness, which corresponds to the ultimate screening by NOs of the electric field generated by the charged patterns [24]. Consequently, the resulting first assembly only partially screens the electric field, which remains strong enough to attract, on top of the first level of NOs, a second type of NO. After a last drying, a selective combinatorial pattern featuring vertically stacked multilayers of two types of NOs is thus obtained.

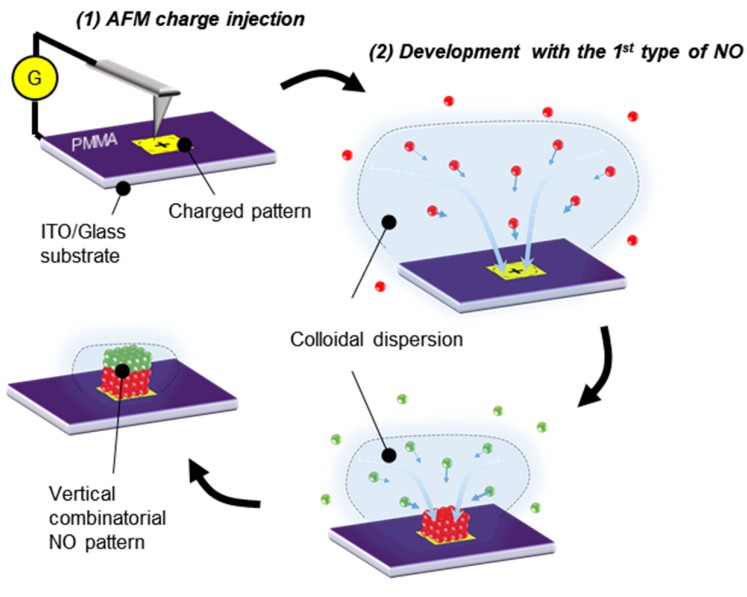

**Figure 1.** Schematics of the principle of vertical combinatorial patterning of colloidal nano-objects (NOs) on surfaces by AFM nanoxerography, here using two types of NOs (colored in red and green). The blue-colored region symbolized is the electric field extension generated by the charged pattern. This region is reduced over time due to the screening of the charged pattern by the successive directed NO assemblies.

To make the proof of principle of this combinatorial NO assembling method, we have selected 27 nm × 17 nm × 8 nm and 27 nm × 13 nm × 4 nm quantum nanoplatelets (NPs) with different sulfur content in the core, CdSe(S)/CdZnS, dispersed in octane emitting in the red and green wavelengths, respectively. Their chemical synthesis was adapted by the Nexdot company on the basis of previously published protocols to be compati-

ble with nanoxerography [25,26]. Our previous work demonstrated that these types of NOs are very sensitive to dielectrophoretic forces and can be efficiently and selectively assembled in multilayered patterns using nanoxerography [24]. Typical scanning electron microscopy (SEM) images along with emission and absorption spectra obtained from these two colloidal dispersions are shown in Figure 2. Absorption spectra were recorded on a Cary 60 UV-visible single-beam spectrometer (Agilent Technologies, CA, USA), while photoluminescence (PL) spectra were obtained with a FP 8500spectrofluorometer (JASCO, Tokyo, Japan), with $\lambda_{exc}$ = 350 nm. The photoluminescence quantum yield (PLQY) of the NPs in dispersion was determined (with a 5% error bar) by using the Quantaurus-QYPlus absolute photoluminescence quantum yield C13534 spectrometer (Hamamatsu, Japan) equipped with an integrating sphere, with $\lambda_{exc}$ = 460 nm.

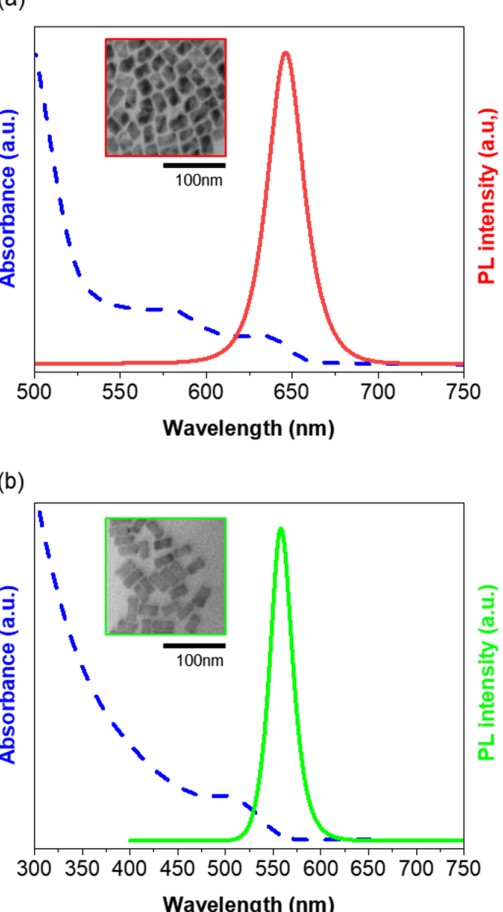

**Figure 2.** Typical SEM image and absorption/emission spectra of (**a**) the colloidal core/shell red-emitting quantum nanoplatelets (peak emission = 646 nm, FWHM = 25 nm, and PLQY (460 nm)~88%) and (**b**) the green-emitting quantum nanoplatelets (peak emission = 558 nm, FWHM = 30 nm, and PLQY (460 nm)~80%).

For sake of simplicity, these two types of NPs are thereafter referred to, respectively, as red, and green NPs.

In a typical experiment, a 20 μm × 20 μm positively charged square was realized within 7 minutes on a 200 nm polymethylmethacrylate (PMMA) spincoated on a 370 nm indium tin oxide (ITO)/1.1 mm glass substrate by controlling the motion of a polarized AFM conductive tip. Positive voltage pulses of 1 ms were applied between the AFM tip and the ITO counter electrode at 50 Hz, with voltage amplitudes ranging from +40 to +80 V. PMMA acts here as the electret layer, featuring excellent charge retention properties [27]. An amount of 20 μL of red and green NPs with concentrations of 1.2–3 × 10$^{12}$ NPs·mL$^{-1}$

and $6 \times 10^{13}$ NPs·mL$^{-1}$, were drop-cast sequentially from 1 to 20 s with a drying step in between. The surface potential of the patterns was measured at any stage by Amplitude Modulation-Kelvin Force Microscopy (AM-KFM) in lift mode (lift height of 30 nm).

### 2.2. Topographical and Optical Characterizations of NP Co-Assemblies

The thickness of each NP level of the vertical combinatorial patterns was measured by AFM in tapping mode after each corresponding development step. Photoluminescence (PL) responses from NP patterns under excitation at a wavelength of 455 nm were collected on an inverted IX73 microscope (Olympus, Japan) coupled an OCEAN-HDX-VIS-NIR spectrometer (Ocean Insigh, FL, USA) and to a grasshopper3 USB3 163 FPS color camera (FLIR, OR, USA) for wide-field acquisitions.

### 2.3. Simulations of the Optical Emission of the Vertical Combinatorial NP Patterns

A numerical electromagnetic tool based on the Green Dyadic Method (GDM) and more specifically the pyGDM toolkit were used to model the optical emission of the vertical combinatorial NP patterns [28,29]. TheseNP patterns were modeled as single disks of 1 μm diameter and 100 nm height in order to maintain a reasonable number of dipoles and simulation time. Each disk was discretized in a mesh of 20 nm cubic cells and was associated with a relatively high real refractive index of 2.5 as a first approximation [30]. The NP emission was modeled by introducing a dipolar source emitter (a point-like electric dipole) inside the disk, in the center. The vertical position and the emission wavelength of the dipolar source were then modified to model the position and the nature (red or green NP-based) of each NP level within the stack.

## 3. Results

### 3.1. Proof of Concept of NP Vertical Co-Assembly

Figure 3 presents typical topographical, surface potential, and optical characterizations of a vertical combinatorial NP patterning at the various protocol stages. In this experiment, a 20 μm charged square was written with +60 V voltage pulses. Using these charging conditions, the PMMA surface was spotless, neither deteriorated nor structured, unlike the case of electrostatic nanolithography of PMMA [31]. The surface potential of the charged pattern measured by AM-KFM equaled approximately 6.3 V. After a first development step of 15 s of the charged sample using a drop of red NPs, the charged pattern area was characterized. It clearly reveals that red NPs were selectively trapped on the charged pattern, retaining the original lateral dimensions of the charged square. The assembly thickness measured by AFM was around 33 nm, confirming a multilayered (2–3 layers) deposition of red NPs. The edges of the red NP pattern appear slightly thicker because of the local increase in the dielectrophoretic forces in these areas. Indeed, their amplitude and therefore attraction were proportional to the electric field gradient to the square [32]. The recorded PL spectrum revealed a main red emissioncentered around 645 nm, in accordance with the PL mapping. The surface potential of the charged pattern partially screened by the red NP assembly was 3.7 V. Once dried, the NPs formed a robust assembly, firmly attached to the surface, which would not diffuse back into the dispersion. After a second development step of 15 s using green NPs, the same sample area was observed again. The thickness of the assembly had doubled, and there was still a 0.9 V surface potential emanating from the pattern. The associated PL spectrum showed this time two emission peaks, centered around 555 and 645 nm, which demonstrated both red and green NP emissions, with a global photoluminescent mapping turning yellow. It is worth noting that these results would have been identical on negatively charged patterns since such NPs are mainly attracted by dielectrophoretic interactions, which are independent of the sign of the injected charges [33].

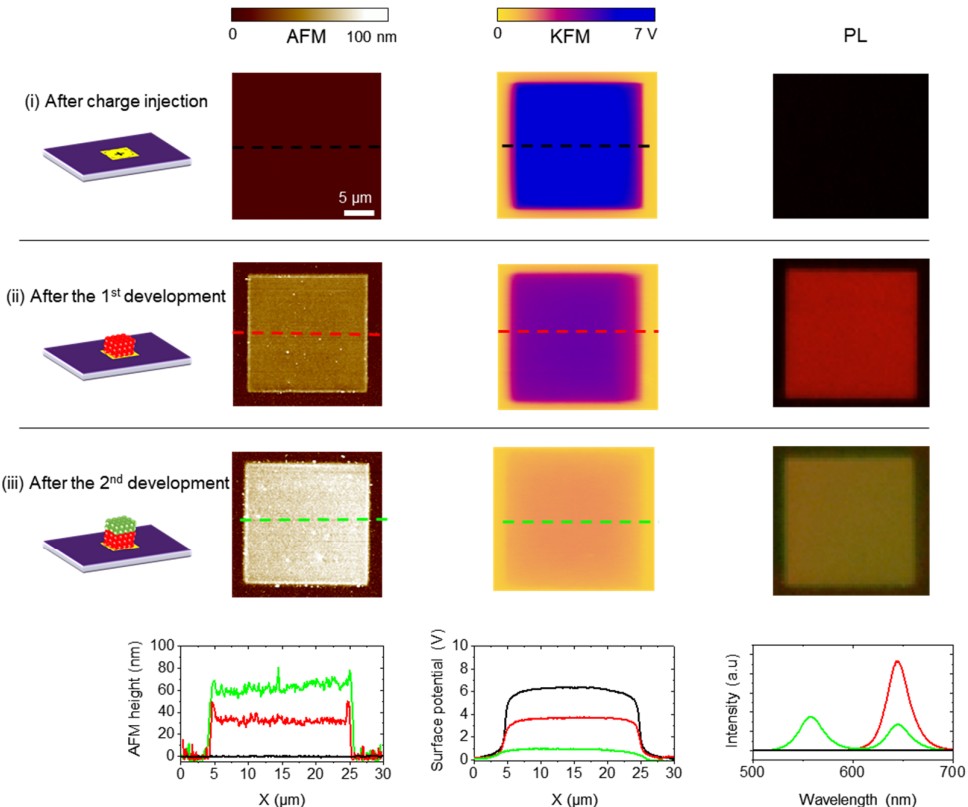

**Figure 3.** Atomic force microscopy (AFM), Kelvin Force Microscopy (KFM), and photoluminescence (PL) mappings at various stages of a vertical combinatorial assembly by AFM nanoxerography of red and green NPs on a 20 µm charged square pattern. The lateral scale is identical for all pictures. The corresponding AFM/KFM profiles and the PL spectra are reported in various colors depending on the protocol step: black after charge injection, red after the first assembly of red NPs, and green after the second assembly of green NPs.

### 3.2. Applications for Dual Colored Patterns

3.2.1. Tuning the Thickness of Each NP Level

To adjust the global PL emission wavelength of the vertical combinatorial NP pattern, we investigate the capability of independently tuning the thickness of each NP level. For that, we varied three key parameters: the surface potential of the charged pattern, the concentration of the NP dispersion and the development time of the charged sample in the NP dispersions.

We previously demonstrated that the time evolution of the thickness of nanoparticle assemblies constructed in multilayers by nanoxerography follows a bell-shaped curve [34,35]. The extremum of this curve, corresponding to the maximum reachable assembly thickness, is reached at a time $t_{opt}$ when the dielectrophoretic forces are no longer predominant over diffusion because of nanoparticles screening the charged pattern. Consequently, the increase in the contact time during the first drop casting, up to the $t_{opt}$ limit without reaching it, will increase the thickness of the first assembly. The closer we are to $t_{opt}$, the thinner the second assembly will be. When the concentration of the dispersion increases, then $t_{opt}$ decreases and the colloidal assembly gets thicker, in good agreement with previous results. However, too high nanoparticle concentrations will lead to too fast and fragile assembly construction and, therefore, lower global assembly thicknesses [24].

Figure 4a reports the respective thicknesses of both red and green NP levels in vertical combinatorial multilayered patterns obtained by AFM nanoxerography in various conditions. The concentration of the used green NP dispersion for the second assembly was fixed at $1.2 \times 10^{12}$ NPs.mL$^{-1}$, while the surface potential of the charged pattern and

the concentration of the red NP dispersion used for the first assembly were varied. The development time for both dispersions was fixed at 15 s.

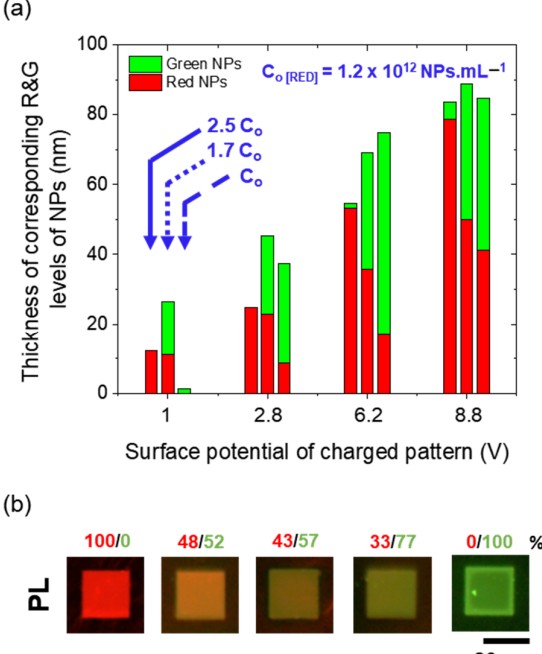

**Figure 4.** (**a**) Evolution of the thicknesses of the red and green NP levels forming vertical combinatorial patterns as a function of the initial surface potential of the charged pattern and the concentration of the red NP dispersion. (**b**) PL images of five red and green NP combinatorial 20 μm square patterns. The different compositions of each NP combinatorial pattern (stated in percentages of the total NP pattern thickness) lead to different colors.

Experimental results confirmed that increasing the surface potential of the charged pattern leads to a thicker first red NP level. An increase of the concentration of the red NP dispersion for a fixed surface potential also results in a thicker first red level. Regarding the second green NP level, we observe that for a given charged pattern, it is thicker for a thinner first red NP level. Indeed, depending on how strong the first NP assembly screens the charged pattern for a given surface potential (i.e., how thick the first assembly is), the second one is building up accordingly based on the remaining attractive electrostatic electric field. In summary, tuning these three parameters allows for the construction of vertical combinatorial assemblies of controlled thicknesses featuring various contributions (i.e., thickness of each NP-multilayered level) from red and green NPs. Consequently, it is possible to access different colors for vertical combinatorial NP patterns ranging from red to green (cf. Figure 4b).

### 3.2.2. Flipping the Sample

By simply flipping the samples (looking through the substrate/assembly), the vertical combinatorial NP patterns present an extra-specific color. To illustrate this phenomenon, some of the various vertical combinatorial assemblies studied in the previous section were optically characterized. We selected NP patterns labelled from 1 to 4 with various compositions of red/green NPs, respectively 0/100, 23/77, 25/75, 48/55 (stated in percentages of the total NP pattern thickness). PL images and spectra of vertical combinatorial assemblies were first recorded looking at the sample from the bottom, through the transparent ITO/glass substrate (Figure 5a).

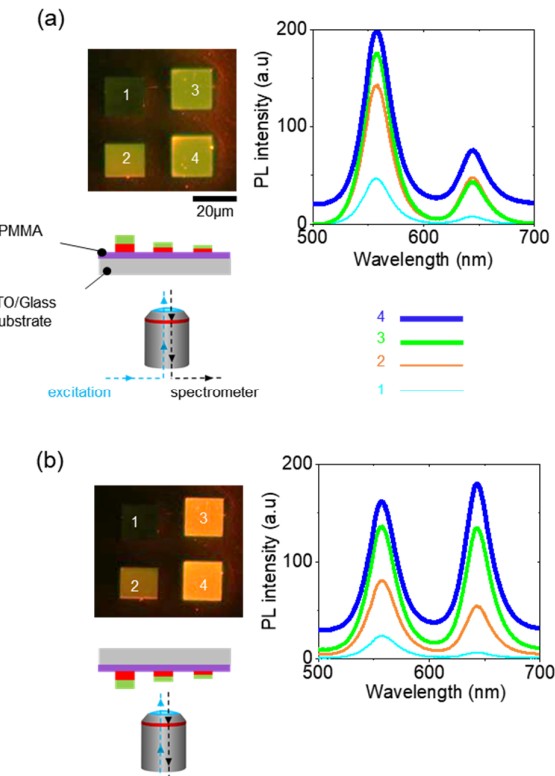

**Figure 5.** PL mapping and corresponding spectra of four co-assembled squares (initially charged with voltage pulses ranging from +40 V to +80 V, from 1 to 4), composed of a first red NP level with a second green NP level on top, where measurements were made from (**a**) the substrate side and (**b**) the assembly side. The scale bar is identical for both sets of PL images.

We observe that the position of the red and green peaks has not shifted from one square to another; only their amplitude has evolved. It was confirmed that decreasing the surface potential of the charged squares (from squares labeled 4 to 1) reduced the thickness of the first red NP level and, consequently, the amplitude of the PL spectra of the corresponding peak. On the other side, the amplitude of the PL spectra corresponding to the green NP level slightly increases with the surface potential of the charged patterns. It must be noted that a straight comparison and correspondence between the PL spectra and thickness of both colloidal systems was not possible since they do not have the same quantum yield and concentration. Most interestingly, flipping the sample (looking directly at co-assemblies) to run the same characterizations led to different colors for PL mappings and therefore to different emission spectra (Figure 5b). Indeed, it appeared that the red NPs emitted more photons towards the "assembly side" and less towards the "substrate side" while the green QDs emitted more photons towards the "substrate side" and less towards "the assembly side".

## 4. Discussion

Simulations were performed to gain insight into this color changing phenomenon (see details in the Materials and Methods Section) depending on the sample side. Figure 6a illustrates a typical simulation case in which a dipolar source emitting in the red region at a wavelength $\lambda$ = 645 nm was placed at the position z = 30 nm, mimicking a first red NP level in a 100 nm-thick vertical combinatorial NP pattern. The light emitted by the dipolar source was then computed in the Fourier plane corresponding to the upper half-space in air (z >> 0 "NP assembly side") (Figure 6b top). The dashed red circle corresponds to the maximal angle to collect light with a 0.7 numerical aperture microscope objective, as was the case in the experiments. It turns out that all the light emitted in the upper half-space in air was experimentally collected by the microscope objective in this case. Figure 6b bottom

shows the Fourier plane of the emitted light within the lower half-space in glass (z < 0 "substrate side"). The presence of the glass substrate reduces the angle of collection from 44.4° in air to 27.8° in glass (red dashed circle). The Total Internal Reflection (TIR) angle, above which the emitted light is trapped and cannot be collected experimentally with an objective in air in the lower half space, is indicated by a white dashed circle in Figure 6b bottom. Thus, only a small emission lobe from the center can be collected in this case while two intense emission lobes are outside of the experimental collection solid angle.

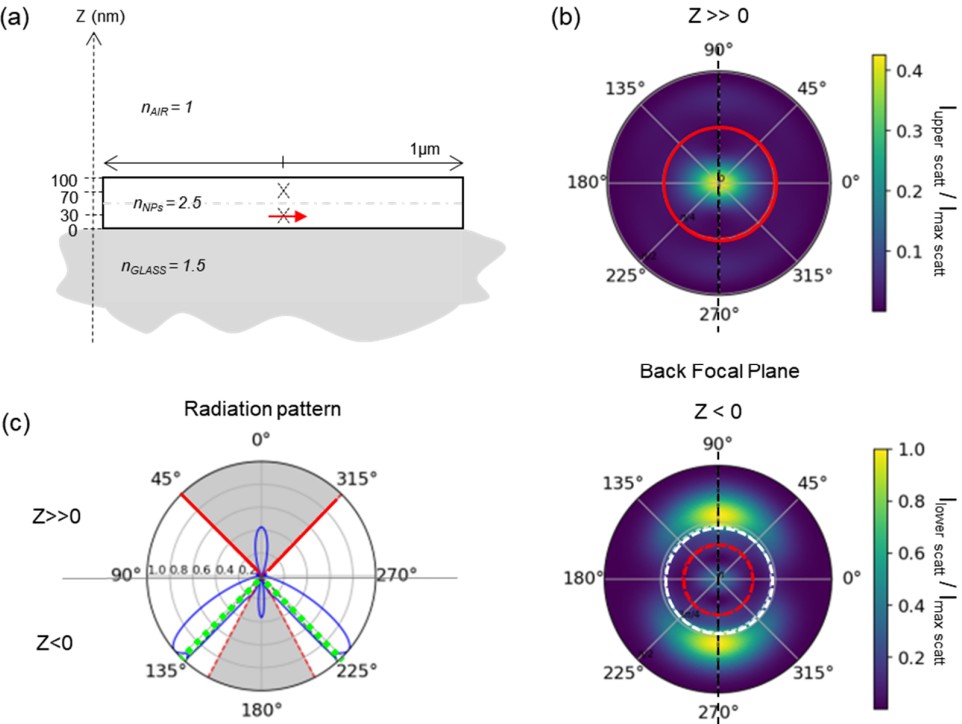

**Figure 6.** (**a**) Schematic crossed section of the NP combinatorial pattern model used for simulations: 100 nm thick disk of $n_{NP} = 2.5$ on top of a glass substrate of $n_{glass} = 1.5$. The glass substrate corresponds to the lower half space. A single dipolar source emitter is placed either at z = 30 nm or 70 nm (on dotted crosses), respectively, in the first and second levels of the NP assembly. In this example, a dipolar source emitting in the red region at $\lambda = 645$ nm (red arrow) is positioned at z = 30 nm, mimicking a first level of red NP. (**b**) Fourier plane of the emitted field, in relative intensity, within (**top**) the upper half space in air (z >> 0), and (**bottom**) the lower half space in glass (z < 0). (**c**) A radiation pattern (blue line), in relative intensity, corresponding to a cross-section along the black dashed line presented in (**b**). The maximal angle of collection of the microscope objective is represented for the upper half space by the red lines (44.4°) and for the lower half space by the red dashed lines (27.8°). The Total Internal Reflection angle represented by the green-dashed line in (**c**) corresponds to the white-dashed line in (**b**) bottom.

Figure 6c presents the radiation pattern corresponding to a cross-section along the black dashed line drawn in Figure 5b. It highlights the two central lobes (θ = 0°) gathered within the collection objective angle, one in each half space. In the lower half space (z < 0), the two large lobes close to the (TIR) angle are not collected. The ratio K corresponding to the collected intensity in the upper half space over the collected intensity in the lower one gave a value of 2.2. It thus points out that there was more collection in the upper half space than in the lower one. If a dipolar source emitting in the green region at a wavelength $\lambda = 555$ nm is now placed in the upper position of the stack (at a position z = 70 nm from the substrate), K turned to 0.6, meaning that there is more collection in the lower half space in that case.

Those simulation results are in good qualitative agreement with the experimental results shown in Figure 5. Indeed, for a charge injection with +70 V pulses, the red (respectively green) peak was almost two times higher (respectively lower) seen from "the NP assembly side" (in air) than the red (respectively green) peak seen from the "substrate side" (in glass).

Since light is preferentially emitted in the medium of a higher index ($n_{NP}$ = 2.5 > $n_{glass}$ = 1.5 > $n_{air}$ = 1), the emission from NPs (red or green) of a given level is steered towards the complementary NP level (green or red, respectively).

Aiming to check the consistency of the model and confirm this assumption, the same simulations were run with reversed geometry, mimicking a second red NP level within the global stack. When adding a red dipolar source emitter at 70 nm, K was equal to 0.9. On the other hand, when adding a green dipolar source emitter at 30 nm, K was equal to 2.8. Therefore, in this inverted configuration, the green NP emission was preferentially collected in the upper direction, whereas the red ones were preferentially collected in the lower direction, demonstrating a symmetrical behavior of the K values. Moreover, changing the diameter of the disk did not affect K, excluding an antenna effect for a given size. Consequently, these simulations prove that the relatively high index of the NP levels is responsible for the partially directive emission observed in the experiments.

## 5. Conclusions

This work gives the proof-of-concept of a vertical co-assembly by nanoxerography using two types of CdSe(S)/CdZnS core/shell quantum nanoplatelets as model colloidal nanoparticles. Red and green NP-multilayered levels of tunable thicknesses were assembled on top of each other onto previously micrometric charged micropatterns thanks to dielectrophoretic forces. We demonstrated that the color of such vertical combinatorial NP micropatterns can be finely adjusted by tuning the thickness of each NP level within the stack through the surface potential of the charged patterns, the concentration of the colloidal dispersion, and/or the development contact times in the two dispersions. The high index of the quantum nanoplatelets finallydrove the resulting photoluminescence emission of the vertical combinatorial patterns, which explains the color change depending on the observation side.

Since nanoxerography is a versatile directed assembly method that could be applied to a wide range of colloids, this work thuspaves the way for other types of original vertical combinatorial micropatterns. Furthermore, while sequential AFM nanoxerography has been selected here to facilitate investigations, these results can be transposed to other existing parallel injection charge techniques (like electrical microcontact printing [36,37] or electrical nanoimprinting [38]) more appropriate to address cm$^2$ areas and to answer industrial constraints for future possible applications.

**Author Contributions:** Conceptualization, G.P., S.R., E.P., R.H. and L.R.; methodology, G.P., E.P., R.H., A.C. and L.R.; software, R.H. and A.C.; validation, G.P., S.R., E.P., A.C. and L.R.; formal analysis, G.P., S.R., E.P., R.H., L.S.M., A.C. and L.R.; investigation, G.P. and R.H.; resources, S.R. and L.S.M.; data curation, G.P., E.P., R.H., A.C. and L.R.; writing—original draft preparation, E.P., R.H. and A.C.; writing—review and editing, G.P., E.P., A.C., L.S.M., M.D. and L.R.; visualization, G.P., E.P., L.S.M., R.H., A.C. and L.R.; supervision, E.P., A.C., M.D. and L.R.; project administration, E.P., A.C., M.D. and L.R.; funding acquisition, E.P., A.C., M.D. and L.R. All authors have read and agreed to the published version of the manuscript.

**Funding:** This research was supported through the grant NanoX n°ANR-17-EURE-0009 in the framework of the "Programme des Investissements d'Avenir" and through the grant n°ANR-20-CE09-0019-01 in the framework of the "Appel à projets génériques 2020".

**Data Availability Statement:** The data presented in this study are available on request from the corresponding author. The data are not publicly available due to privacy reasons.

**Conflicts of Interest:** The authors declare no conflict of interest.

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
