# Peer review of "Electrostatically Driven Vertical Combinatorial Patterning of Colloidal Nano-Objects"

_colloids, doi:10.3390/colloids7010006_

Round 1

Reviewer 1 Report

This work presents a vertical assembly of red/green QD nanoplates through AFM nanoxerography. The authors were able to tune the thickness of two layers by controlling the concentration and surface charge. In the characterization part, the authors emphasized the direction preference of red/green QD emission. The experiments were well-designed under different conditions, and the results were discussed rigorously. At the end of the discussion part, the design of reversing assembly geometry is a better control compared to the flipping detector and would be better if relative data were presented in the paper. I would like to suggest that this paper be published in the journal after minor revisions. I also have the following comments based on the manuscript.

Please demonstrate the core/shell structure of two different nanoplatelets, and also add the description of absorbance spectra in the caption of Figure 2.

Line 98, provide the full name of ITO.

In figure 4, please label the thickness of two layers in the bottom PL images.

In figure 5, four different spectra are hard to identify with thickness, especially No3 and No4. It will be better to change to different colors.

In the “flipping the samples” part, does the PL detection control the observation distance on both the assembly side and subtract side? Does subtract and the neighboring layer of QD absorb or enhance light at red/green wavelength?

Please provide details about simulation experiments. What kind of dipolar source was used? In Figure 6a, what do the two crosses and the dotted dash line in the middle represent? Line 240, please provide the full spelling of “0.7 NA”. 

Reviewer 2 Report

In this paper, the authors propose to use an alternative electrically driven directed assembly technique, called nanoxerography, to create vertical combinatorial NO patterns. This work serves as a proof of concept of vertical combinatorial NO patterns by nanoxerography using two types of highly photoluminescent CdSe(S)/CdZnS core/shell  nanoplatelets as model colloidal nanoparticles, and how it has been applied to make dualcolored patterns. It is well-written and can be published after minor modification.

1. For figure 4, what is the wavelength that the authors use to induce PL?

2. First, a 20 µm x 20 µm charge square 128 was written with a voltage amplitude of +60 V. Why did the authors choose 60 V as the typical voltage?
